# Absence of Surface Water Temperature Trends in Lake Kinneret despite Present Atmospheric Warming: Comparisons with Dead Sea Trends

**Pavel Kishcha [1,\*] , Boris Starobinets [1], Yury Lechinsky [2] and Pinhas Alpert [1]**

1    Department of Geophysics, Tel Aviv University, Tel Aviv 69978, Israel; starob@tauex.tau.ac.il (B.S.); pinhas@tauex.tau.ac.il (P.A.)

2    Kinneret Limnological Laboratory, Israel Oceanographic and Limnological Research, Migdal 1495000, Israel; yuryl@ocean.org.il

\*    Correspondence: pavel@cyclone.tau.ac.il

**Abstract:** This study was carried out using Moderate Resolution Imaging Spectroradiometer (MODIS) 1 km × 1 km resolution records on board Terra and Aqua satellites and in-situ measurements during the period (2003–2019). In spite of the presence of increasing atmospheric warming, in summer when evaporation is maximal, in fresh-water Lake Kinneret, satellite data revealed the absence of surface water temperature (SWT) trends. The absence of SWT trends in the presence of increasing atmospheric warming is an indication of the influence of increasing evaporation on SWT trends. The increasing water cooling, due to the above-mentioned increasing evaporation, compensated for increasing heating of surface water by regional atmospheric warming, resulting in the absence of SWT trends. In contrast to fresh-water Lake Kinneret, in the hypersaline Dead Sea, located ~100 km apart, MODIS records showed an increasing trend of 0.8 °C decade$^{-1}$ in summer SWT during the same study period. The presence of increasing SWT trends in the presence of increasing atmospheric warming is an indication of the absence of steadily increasing evaporation in the Dead Sea. This is supported by a constant drop in Dead Sea water level at the rate of ~1 m/year from year to year during the last 25-year period (1995–2020). In summer, in contrast to satellite measurements, in-situ measurements of near-surface water temperature in Lake Kinneret showed an increasing trend of 0.7 °C decade$^{-1}$.

**Keywords:** freshwater lakes; saline lakes; lake kinneret; dead sea; lake surface water temperature; water temperature trends; lake remote sensing

## 1. Introduction

Lake surface water temperature (SWT) is one of the main factors determining evaporation and, consequently, energy and moisture exchange at the air-water interface. As a consequence of the air being in contact with the lake water surface, SWT is sensitive to atmospheric warming. Air warming in the overlying atmosphere is reflected in lake surface water temperature as well as in water temperature below the surface. This has potential consequences for a broad range of physical and ecological factors such as thermal structure; lake productivity and ecosystems, in accordance with Adrian et al. [1] and Williamson et al. [2]. This highlights the importance of investigating long-term lake surface water temperature trends.

Within the Jordan Rift valley there are two lakes: the hypersaline Dead Sea (a surface area of 605 km$^2$ and a maximal depth of 300 m) and the fresh-water Lake Kinneret (Sea of Galilee) (a surface area of 106 km$^2$ and a maximal depth of 40 m). Earlier research by Kishcha et al. [3] on Dead Sea surface temperature showed increasing surface water temperature trends of ~0.6 °C decade$^{-1}$ in the steadily shrinking Dead Sea. That study was carried out using MODIS data of 5 km × 5 km resolution during the period 2000–2016

on board the NASA Terra satellite [3]. Moreover, in the summer months, in the absence of water mixing and under uniform solar radiation, evaporation was the main causal factor of the observed pronounced spatial heterogeneity in Dead Sea surface temperature, according to [4,5].

Lake Kinneret is located in the northern section of the Jordan Rift valley at 210 m b.s.l. Its length is ~32 km and its width ~12 km. The lake is fed partly by underground springs, but its main source is the Jordan River, which flows through it from north to south and exits the lake at the Degania Dam. The purpose of the dam is to regulate water levels in the lake and water flow from the lake into the lower Jordan River. The dam is opened in the case when coastal areas are threatened by flooding. The climate over the Kinneret area is characterized by the high annual averaged air temperature equal to 21 °C and maximal summer air temperature exceeding 36 °C [6]. There is no precipitation in the hot season (June–September), while rainfall of approximately 400 mm takes place during the rainy season from December to April [6]. Strong winds exceeding 10 m/s (with gusts up to 30 m/s) are regularly observed over Lake Kinneret in early afternoon in the summer months. Such strong winds are caused by the penetration of Mediterranean Sea breezes into the lake area [7–11]. During summer mornings, from ~07:00 LT to ~11:00 LT (local time), Kinneret lake breezes develop, blowing from the lake towards the shores [8]. Gal et al. [12] analyzed year-to-year variations of the annual maximum water temperature, using buoy measurements of water temperature at a depth of 1 m. They found a long-term (1970–2019) trend of increasing water temperature (at the depth of 1 m) at a rate of 0.4 °C decade$^{-1}$ [12]. This rate was higher than that of 0.28 °C decade$^{-1}$ reported for the mean temperature of the upper 1-m water layer (epilimnium) over the period 1969–2008 [13]. The authors of [12,13] considered that atmospheric warming was responsible for the above-mentioned water temperature trend.

We investigated long-term trends of SWT in Lake Kinneret using both satellite and in-situ measurements. This was carried out using the 17-year period of satellite-based MODIS 1 km × 1 km resolution records (2003–2019). The satellite-based trends in lake surface water temperature were compared with those based on in-situ measurements conducted during the same 17-year period. MODIS SWT trends in fresh-water Lake Kinneret were compared with those in the hypersaline Dead Sea. The two lakes are located in similar summer environmental settings.

## 2. Materials and Methods

We investigated long-term trends of surface water temperature in Lake Kinneret focusing on the region (32.67°N–32.93°N, 35.46°E–35.71°E) covering the lake and surrounding land areas (Figure 1). This lake is flanked by hills of ~400 m height: the Galilee Hills to the west and the Golan Heights to the east. North of Kinneret Lake is Mount Hermon of 2800 m height. South of Kinneret Lake is the Dead Sea (Figure 1a).

For the monthly mean Lake Kinneret surface water temperatures, we used Collection-6 (C6) of the following MODIS (Moderate Resolution Imaging Spectroradiometer) Land Surface Temperature (LST) products during the period from 2003 to 2019: MOD11A1 [14] and MYD11A1 [15]. The MOD11A1 C6 product provides daily per-pixel MODIS LST Level-3 data from the Terra satellite at $1 \times 1$ km$^2$ spatial resolution in the daytime at approximately 10:30 LT and in the nighttime at 22:30 LT. The MYD11A1 C6 product provides daily per-pixel MODIS LST Level-3 data from the Aqua satellite at $1 \times 1$ km$^2$ spatial resolution in the daytime at approximately 13:30 LT and in the nighttime at 01:30 LT. MODIS measures lake surface water temperature in the skin layer of 10–20 μm, which is in direct exposure to air [16,17]. Wan [18] conducted validation of MODIS/Terra and MODIS/Aqua LST products at 42 sites around the globe (including the Mediterranean region), in different seasons and years. He showed that the mean MODIS Collection-6 LST product error is within ±0.6 °C.

To accurately investigate trends in Kinneret SWT, it was essential to exclude MODIS LST pixels with land contamination. This was carried out using the water body identifier

dataset by Carrea et al. [19]. Being based on Envisat satellite measurements of backscattered intensity locating water bodies with horizontal resolution of 300 × 300 m, this dataset allowed us to discriminate accurately between water and land areas.

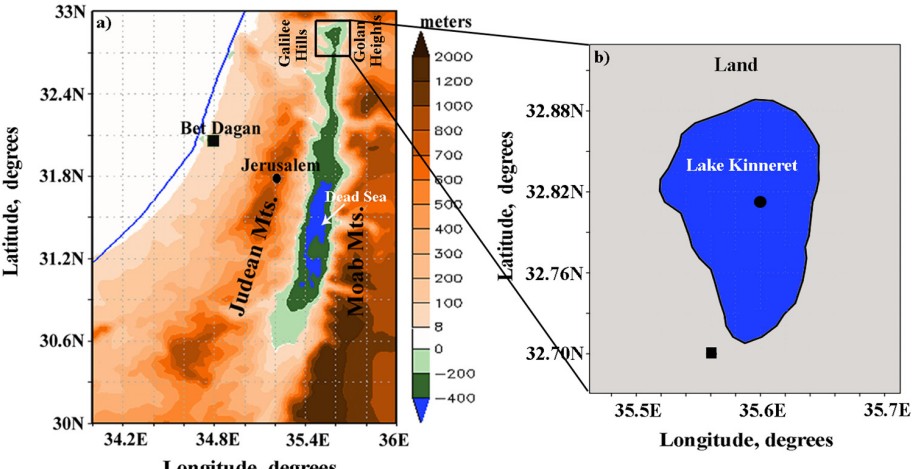

**Figure 1.** (**a**) Topographic map of the south-east Mediterranean region, and (**b**) geographic map of the region under study including Lake Kinneret and surrounding land areas. The black circle shows the location of ship measurements of water temperature (32.81° N, 35.60° E), while the black square shows the location of the Zemah meteorological station (32.70° N, 35.58° E).

Similarly to Kishcha et al. [3], in order to obtain long-term trends of Kinneret SWT, the above-mentioned satellite monthly data averaged over Lake Kinneret (Figure 1b) were deseasonalized by removing 17-year averages from any given month. For the obtained anomalies, the slope of a linear fit was used to determine Kinneret SWT trends during the 17-year period under investigation (2003–2019). To estimate the significance level (*p*) value of surface temperature trends, normally distributed residuals of the linear fit were used in a t test [20,21]. The obtained *p* values less than 0.05 correspond to statistically significant surface temperature trends at the 95% confidence level. In the current study, the slope, its standard error, and significance level (*p*) of the linear fit were obtained using a linear regression approach (Tables A1–A3). This was carried out using OriginLab software from OriginLab Corporation, Northampton, MA, USA [22].

The obtained long-term SWT trends, based on satellite data, were compared with in situ (shipboard) measurements of near-surface water temperature. To this end, weekly shipboard measurements of water temperature profiles were used during the study period, between 09:00 LT–10:00 LT, at the deepest point of ~40 m near the lake center (Figure 1). Shipboard measurements of near-surface water temperature were taken at a depth of 0.1 m using the AML MINOS X probe [23]. In contrast to satellite measurements, these in-situ measurements were conducted in the water layer, which was not in direct exposure to air. We also used in-situ, shipboard, measurements of water temperature at a depth of 1 m and of 2 m.

To study the effect of atmospheric factors on long-term trends in Kinneret SWT, we used available 10-min measurements of 2-m air temperature, surface solar radiation, and near-surface wind speed, during the study period from 2003 to 2019 [24]. These measurements were taken at the Zemah meteorological station, located in the vicinity of Lake Kinneret (Figure 1), during the same time as the in-situ, shipboard, water temperature measurements (09:00 LT–10:00 LT). In addition, we analyzed yearly data of Kinneret water levels, based on available measurements from 1935 to 2020 (Table A4). These data of Kinneret water levels were used for the analysis of climate-related long-term changes in the lake.

For the purpose of comparison, the obtained long-term MODIS SWT trends in freshwater Lake Kinneret were compared with those in the hypersaline Dead Sea. The two lakes are ~100 km apart, located in similar summer environmental settings.

## 3. Results

### 3.1. Long-Term Changes in Kinneret Water Levels

Measured year-to-year variations in Kinneret water levels from 1935 to 2020 revealed a decrease, in accordance with the obtained linear fit (Figure 2a). According to previous estimates, there are two possible factors for this decrease: (1) reduction of ground water recharge (following rainfall events) by an average rate of ~3 $10^6$ m$^3$ year$^{-1}$ and (2) increase in evaporation by the rate of ~0.4 $10^6$ m$^3$ year$^{-1}$ [13,25]. Hence, the main reason for the decrease in Kinneret water levels is a decrease in precipitation.

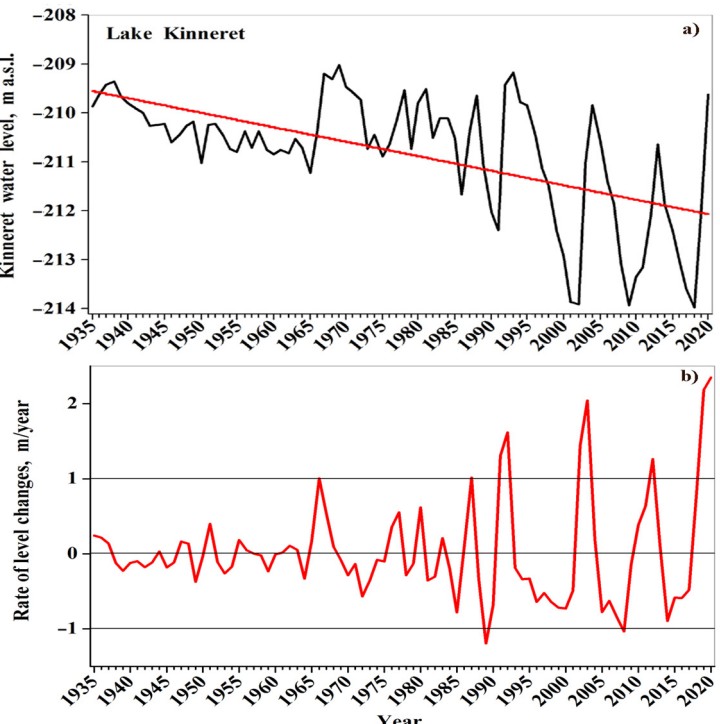

**Figure 2.** (**a**) Yearly data of the Kinneret water levels from 1935 to 2016; and (**b**) the rate of Kinneret water level changes from year to year. The straight red line designates a linear fit.

Moreover, one can see an essential difference between the following three periods: (1) before the mid-1960s, (2) from the mid-1960s to the mid-1980s, and (3) after the mid-1980s (Figure 2a). We analyzed a rate of year-to-year variations in Kinneret water levels during the period from 1935 to 2020 (Figure 2b). This rate was estimated as the difference between the measured water level in the given year and that in the previous year. One can see that, before the 1960s, the obtained rate of year-to-year variations in Kinneret water levels was minimal; from the mid-1960s to the mid-1980s it was up to 1 m year$^{-1}$; while after the mid-1980s, it was up to 2 m year$^{-1}$ (Figure 2b). This fact clearly indicated the presence of significant precipitation-related changes after the 1980s due to the rotation of dry years and rainy years. Note that Lake Kinneret supplies about 30% of the national water demand, according to Sade et al. [25]. This anthropogenic factor (such as an increase in freshwater demand) was superimposed on the climate-related changes after the 1980s [26]. After 1990, the rate of water level change revealed significant variations (Figure 2b). Such significant interannual variations were mainly caused by precipitation changes but not by changes in freshwater demand. Therefore, the period from 2003 to 2019 is mainly characterized by significant fluctuations in Kinneret water levels due to precipitation changes.

### 3.2. Long-Term Water Temperature Trends Based on In-Situ Measurements for All Months

Available in situ measurements were conducted at the deepest point of ~40 m near the lake center (Figure 1). At this deepest point, the influence of the bottom temperature on the surface water temperature is minimal. The in-situ month-to-month measurements showed the statistically significant increasing trend of 0.6 °C decade$^{-1}$ in near-surface water temperature, during the study period. (Figure 3a,b; and Table A1).

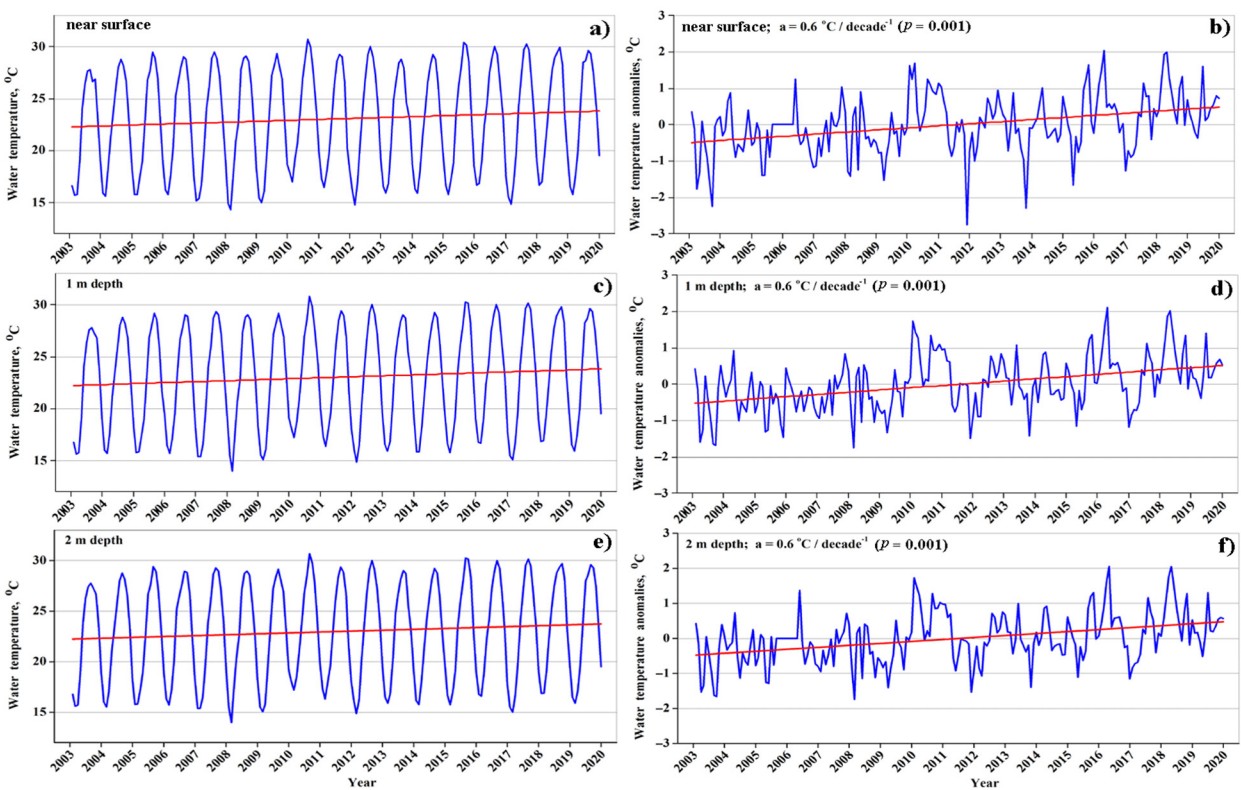

**Figure 3.** Month-to-month variations of (**a**,**b**) Kinneret near-surface water temperature; water temperature at (**c**,**d**) a depth of 1 m and at (**e**,**f**) a depth of 2 m; based on in situ measurements, during the 17-year period under study. The left column (**a**,**c**,**e**)represents original monthly data while the right column (**b**,**d**,**f**) represents their associated deseasonalized monthly anomalies. The straight red lines designate linear fits.

The same trends of 0.6 °C decade$^{-1}$ were found in water temperature at a depth of 1 m, and at a depth of 2 m (Figure 3c–f; and Table A1). We also found a high correlation of over 0.93 in corresponding month-to-month water temperature anomalies between near-surface water temperature and water temperature at a depth of 1 m (Figure 3b,d). Furthermore, a high correlation of over 0.95 was found in the corresponding month-to-month water temperature anomalies between near-surface water temperature and water temperature at a depth of 2 m. The above high correlation indicates that the same causal factor (such as vertical water mixing) was responsible for the observed time variations of both near-surface water temperature and water temperature at a depth of 1 m and of 2 m.

To study possible effects of climatic factors on trends in near-surface water temperature in Lake Kinneret, based on in-situ measurements, we used available 10-min meteorological measurements of (1) 2-m air temperature; (2) near-surface wind speed; and (3) surface solar radiation. These meteorological measurements were taken at the Zemah meteorological station, during the same time as in-situ water temperature measurements (09:00 LT–10:00 LT). 10-min meteorological measurements between 09:00 LT and 10:00 LT were monthly averaged during the 17-year study period (2003–2019) (Figure 4).

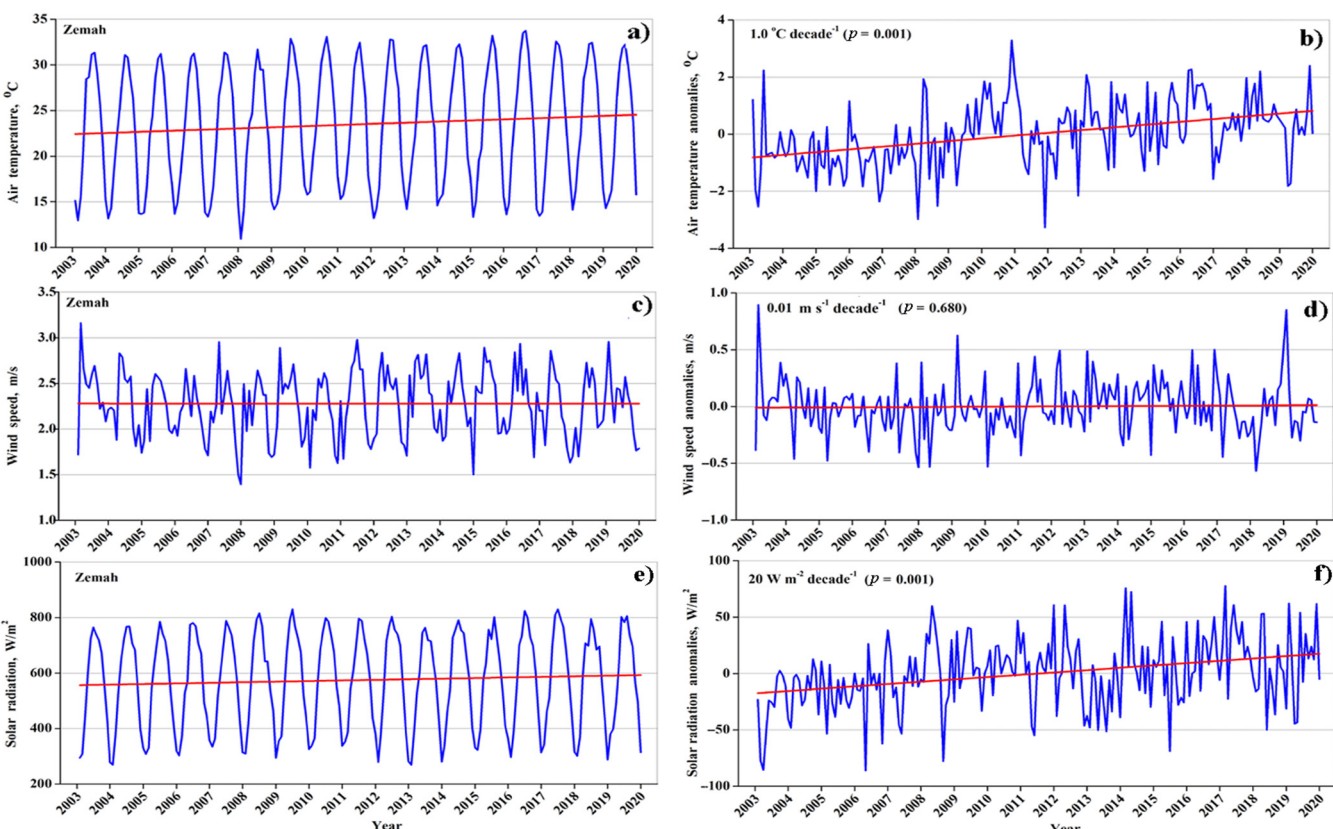

**Figure 4.** Month-to-month variations of (**top panel**) (**a**,**b**) 2-m air temperature, (**middle panel**) (**c**,**d**) near-surface wind speed, and (**bottom panel**) (**e**,**f**) surface solar radiation taken at the Zemah meteorological station, located in the vicinity of Lake Kinneret, during the 17-year period under study. The left column (**a**,**c**,**e**) represents original monthly data while the right column (**b**,**d**,**f**) represents their associated deseasonalized monthly anomalies. The straight red lines designate linear fits.

Month-to-month meteorological measurements of air temperature showed a strong increasing trend of 1 °C decade$^{-1}$, indicating the presence of significant regional atmospheric warming over Lake Kinneret during the study period (Figure 4a,b and Table A2). During the study period, meteorological monthly data showed no statistically significant trends in wind speed (Figure 4c,d). Month-to-month pyranometer measurements during the study period showed a statistically significant increasing trend of 20 W m$^{-2}$ decade$^{-1}$ in surface solar radiation (Figure 4e,f). This increasing trend in solar radiation can be explained by a decrease in cloud cover, in accordance with Paudel et al. [27].

Because the absence of any trend in wind speeds, the observed trends in Kinneret near-surface water temperature (based on in-situ measurements) cannot be explained by long-term changes in wind speed. Increasing atmospheric warming and increasing trends in solar radiation could be the climatic factors responsible for the obtained increasing long-term trends in near-surface water temperature.

### 3.3. Trends in Daytime MODIS-Based SWT in Lake Kinneret for All Months

MODIS measures lake surface water temperature in the skin layer of 10–20 μm which is in direct exposure to air. Therefore, satellite data are sensitive to the effect of evaporation on surface water temperature, in contrast to in-situ measurements. The comparison between in-situ and satellite measurements of SWT is of importance, as it could provide us with some insight into the contribution of evaporation to SWT trends.

MODIS satellite data of skin surface temperature, during the study period, were averaged over the specified Kinneret water area. The obtained time-series of MODIS surface water temperature in Lake Kinneret showed a statistically significant increasing

SWT trend of 0.5 °C decade$^{-1}$ at 10:30 LT, based on MODIS/Terra data, and 0.4 °C decade$^{-1}$ at 13:30 LT, based on MODIS/Aqua data (Figure 5b,d and Table A3). The above-mentioned trends in surface water temperature, based on MODIS satellite data (Figure 5), were lower than those based on in-situ measurements (0.6 °C decade$^{-1}$) (Figure 3).

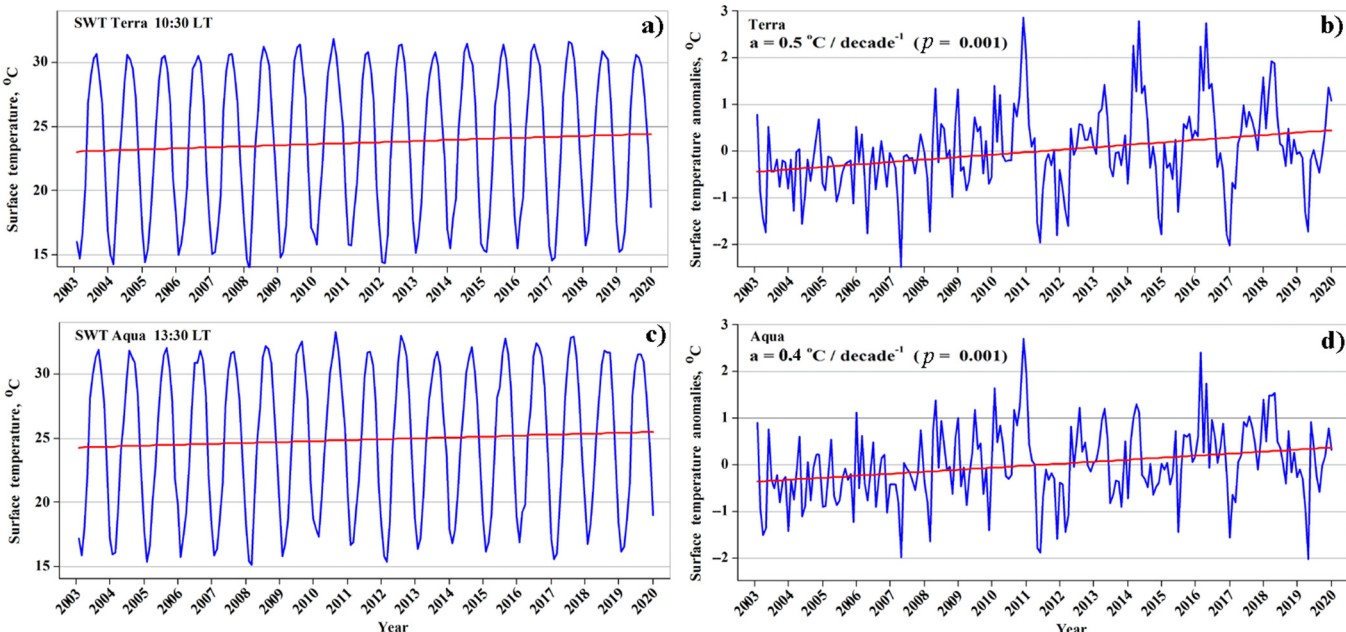

**Figure 5.** Monthly variations of daytime Kinneret SWT during the 17-year period under study. The left column represents original (**a**) MODIS/Terra (10:30 LT) and (**c**) MODIS/Aqua (13:30 LT) monthly data (averaged over the specified Kinneret water area), while the right column (**b**,**d**) represents their associated deseasonalized monthly anomalies. The straight red lines designate linear fits.

Month-to-month variations of daytime MODIS surface water temperature were not homogeneous during the period under study from 2003 to 2019. Specifically, in accordance with the time series of deseasonalized monthly anomalies of MODIS SWT, their deviations from the obtained linear fit were more pronounced after the year 2010 than before (Figure 5b,d). To illustrate this fact, we analyzed variations of absolute values of residuals (with respect to the obtained linear fit) of MODIS/Terra SWT and MODIS/Aqua SWT, during the study period (Figure 6a,b). One can see that the absolute values of residuals of MODIS/Terra SWT before 2010 were mainly lower than 2 °C, while after 2010 they frequently exceeded 2 °C (Figure 6a). A similar phenomenon can be seen in the absolute values of residuals of MODIS/Aqua SWT (Figure 6b).

### 3.4. In-Situ Based Trends in Near-Surface Water Temperature in the Summer Months

As mentioned, increasing atmospheric warming contributed to both in-situ measured near-surface water temperature trends and satellite-based SWT trends. In-situ measurements are insensitive to water cooling due to evaporation, while satellite measurements are sensitive to this cooling. We focused on the summer months for the following reasons: (1) evaporation is maximal; (2) precipitation does not occur; (3) cloud cover is minimal and cannot significantly influence solar radiation.

In situ measurements showed an increasing trend of 0.7 °C decade$^{-1}$ in Kinneret near-surface water temperature in the summer season (Figure 7). Furthermore, in separate summer months (June, July, and August), in-situ measurements showed increasing statistically significant trends in near-surface water temperature (Figure 8a–c).

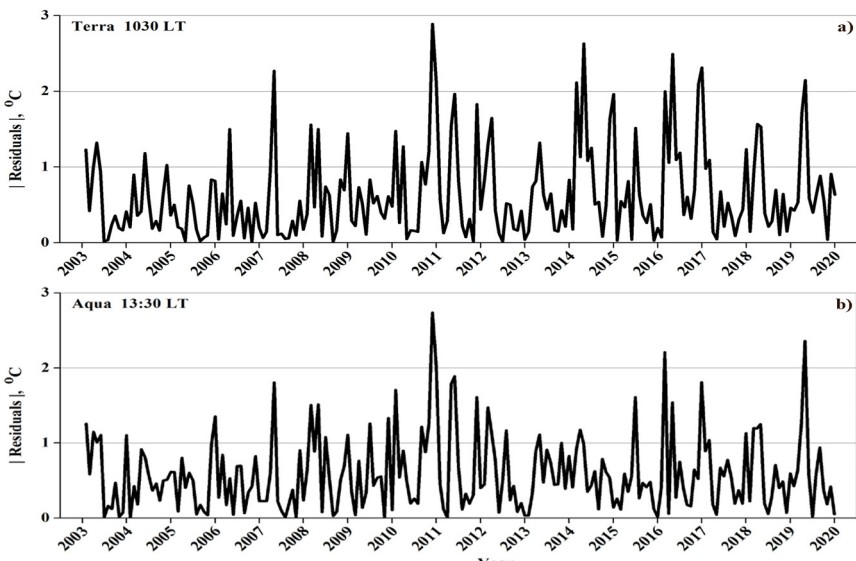

**Figure 6.** Monthly variations of absolute values of residuals with respect to the obtained linear fit of (**a**) MODIS/Terra and (**b**) MODIS/Aqua surface water temperatures.

Increasing atmospheric warming can be a climatic factor responsible for the obtained increasing long-term trends in near-surface water temperature. Indeed, meteorological measurements showed a strong increasing statistically significant trend of 1.1 °C decade$^{-1}$ in air temperature in summer (Figure 9a). Moreover, in each summer month (June, July, and August), the observed increasing statistically significant trends, ranged from 0.9 to 1.3 °C decade$^{-1}$, were observed in air temperature (Figure 10a,d,g; and Table A2). This increasing atmospheric warming contributed to the increasing trends in near-surface water temperature in each summer month (Figure 8). Meteorological measurements showed the absence of wind speed trends in any summer month (Figure 10b,e,h). Pyranometer data showed a weak increasing trend of 22 W/m$^2$ decade$^{-1}$ in surface solar radiation in July, while in June and August there were no statistically significant trends (Figure 10c,f,i).

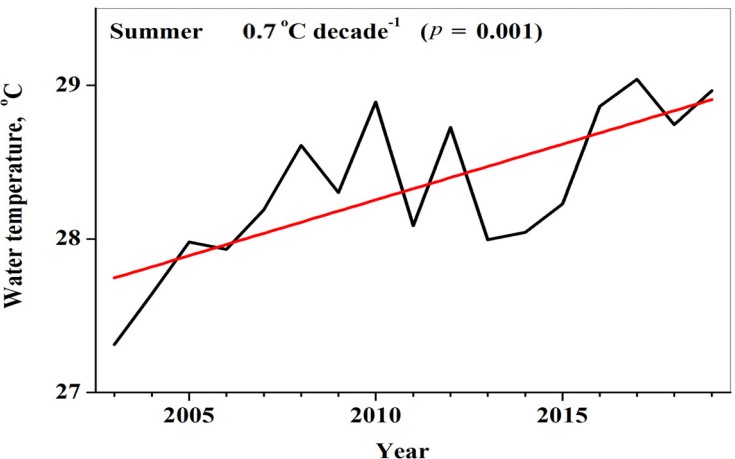

**Figure 7.** Year-to-year variations of Kinneret near-surface water temperature in summer, based on in-situ measurements, during the 17-year period under study (2003–2019). The straight red lines designate linear fits.

*3.5. Insignificant Trends in MODIS-Based SWT in Lake Kinneret in the Summer Months*

In summer, when the effect of maximal evaporation on SWT was significant, a comparison between in-situ and satellite measurements of SWT is of particular importance to get some insight into the contribution of evaporation to SWT trends. In summer, both MODIS/Terra and MODIS/Aqua data showed no statistically significant trends in SWT,

averaged over the whole Kinneret water area (Figure 11a,b). Moreover, both MODIS/Terra and MODIS/Aqua data showed insignificant SWT trends in June, July and August separately (Figure 12 and Table A3). Therefore, in summer, MODIS/Terra data of SWT at 10:30 LT and in-situ measurements of near-surface temperature between 9 LT–10 LT showed different trends (Figure 12a,c,e; and Figure 8a–c).

MODIS records also provide nighttime SWT data at 22:30 LT and 01:30 LT, based on MODIS/Terra and MODIS/Aqua records respectively. Similarly to the daytime SWT trends, nighttime SWT trends were obtained over the whole Kinneret water area. MODIS/Terra data showed statistically insignificant trends in June, July and August separately, at 22:30 LT (Figure 13a,c,e). At 01:30 LT, MODIS/Aqua data showed insignificant trends in July and August (Figure 13d,f), and a statistically significant trend in June (Figure 13b). Thus, in the summer months, not only in the daytime but also at night, satellite data revealed mainly the absence of statistically significant SWT trends, during the 17-year study period. Unfortunately, there were no in-situ measurements of near-surface water temperature in the nighttime.

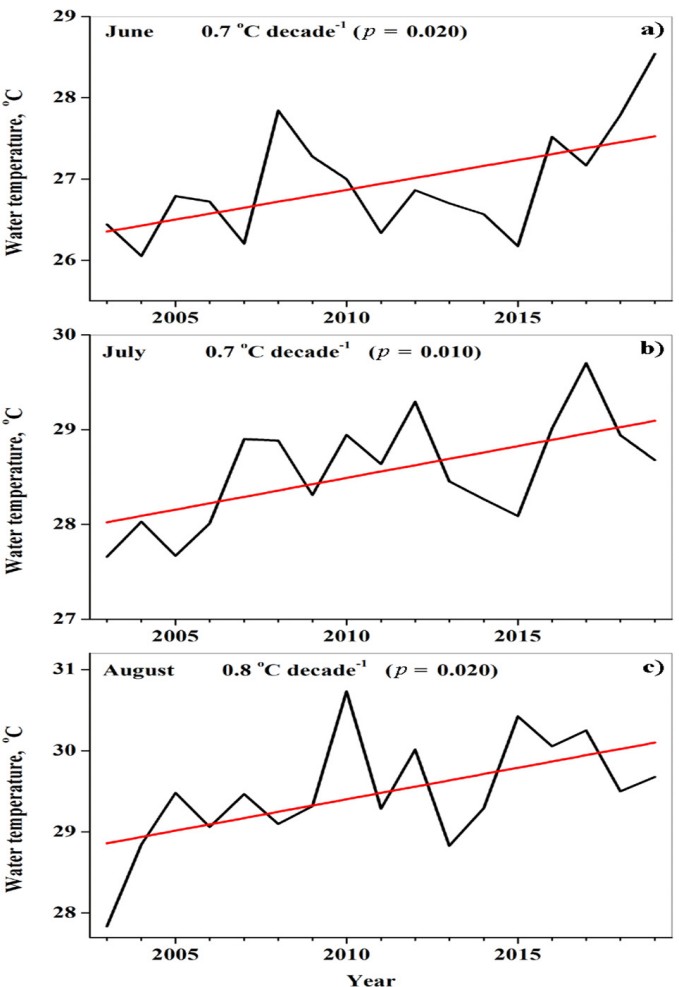

**Figure 8.** Year-to-year variations of Kinneret near-surface water temperature in (**a**) June, (**b**) July, and (**c**) August, based on in-situ measurements, during the period under study (2003–2019). The straight red lines designate linear fits.

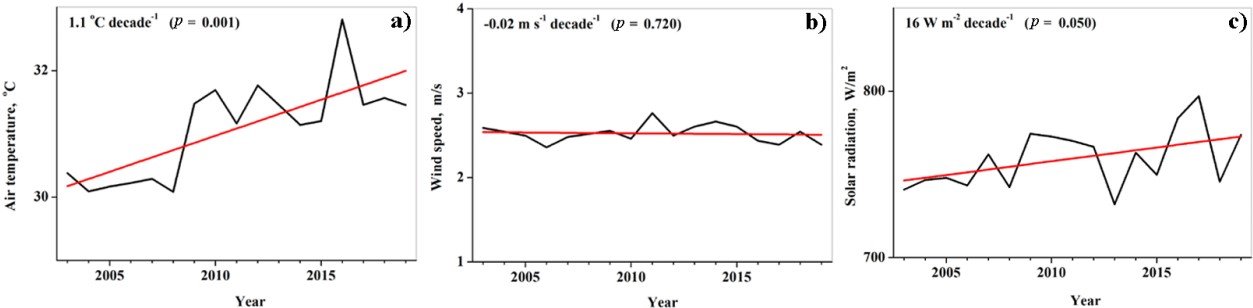

**Figure 9.** Year-to-year variations of (**a**) air temperature, (**b**) wind speed, and (**c**) surface solar radiation measured at the Zemah meteorological station, between 9 LT and 10 LT, in summer (June–August), during the 17-year study period. The straight red lines designate linear fits.

### 3.6. Summer SWT Trends in the Dead Sea

It is worth noting that fresh-water Lake Kinneret and the hypersaline Dead Sea are located at ~100 km apart, in similar summer climatic conditions. These conditions are characterized by the absence of precipitation and by the presence of minimal cloud cover which cannot significantly influence solar radiation. A complex wind regime is observed over the Dead Sea including Dead Sea breezes, the Mediterranean Sea breeze, foehn and local katabatic winds [28–30]. A specific feature of winds over the Dead Sea is strong winds up to 10 m/s in the nighttime (causing significant water mixing), and weak winds of ~2 m/s in the daytime [31]. There was no long-term trend in wind speed either in Lake Kinneret or in the Dead Sea [3]. Despite this fact, satellite MODIS/Terra high resolution data (1 km × 1 km) showed a strong increasing statistically-significant SWT trend of 0.8 °C decade$^{-1}$ in the Dead Sea during the same period (2003–2019), in contrast to that in Lake Kinneret (Figure 14a,b). In accordance with Figure 14, in summer during the study period, MODIS-based Kinneret SWT varied from 30 °C to 31 °C, while Dead Sea SWT steadily rose from 32 °C to 34 °C. Furthermore, satellite MODIS/Terra data showed increasing statistically significant trends in Dead Sea SWT in any separate summer month ranging from 0.7 to 1.0 °C decade$^{-1}$ (Figure 15a–c).

The same linear regression approach was used to estimate the above-mentioned summer trend in fresh-water Lake Kinneret SWT and that in hypersaline Dead Sea SWT. We found essentially different trends in the two lakes. To support our findings based on linear regression, the Mann-Kendall trend test was applied to satellite SWT data in the two lakes [32–34]. This was carried out using XLSTATstatistical software from Addinsoft Inc., New York, NY, USA [35]. Being applied to MODIS/Terra data in summer during the study period (2003–2019), in the Dead Sea, the Mann-Kendall trend test showed a strong increasing SWT trend of 0.9 °C decade$^{-1}$. The obtained $p$ value of 0.001 corresponded to the statistically significant SWT trend at the 95% confidence level. In contrast, in Lake Kinneret, the Mann-Kendall trend test showed a weaker SWT trend of 0.4 °C decade$^{-1}$ which was statistically insignificant ($p$ value was equal to 0.055). Therefore, both the linear regression approach and the Mann-Kendall trend test showed the presence of increasing SWT trend in the hypersaline Dead Sea and the absence of SWT trend in fresh-water Lake Kinneret. Hence, the above mentioned two different trend estimators supported each other.

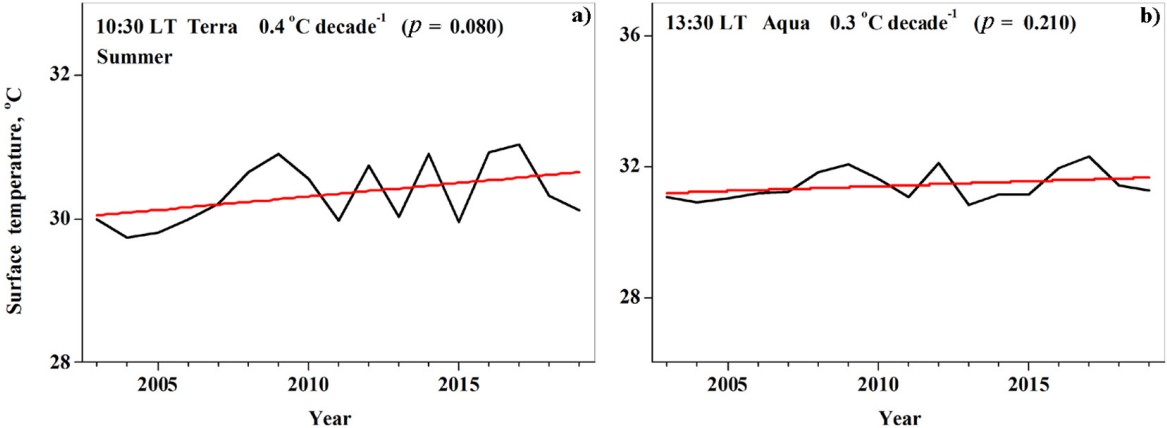

**Figure 10.** Year-to-year variations of (**left column**) (**a,d,g**) air temperature, (**middle column**) (**b,e,h**) wind speed, and (**right column**) (**c,f,i**) solar radiation measured at the Zemah meteorological station, between 9 LT and 10 LT, in the summer months. The top panel (**a–c**) corresponds to June; the middle panel (**d–f**) corresponds to July, and the bottom panel (**g,h,i**) corresponds to August. The straight red lines designate linear fits.

**Figure 11.** Year-to-year variations of Kinneret surface water temperature in summer based on (**a**) MODIS/Terra data at 10:30 LT and on (**b**) MODIS/Aqua data at 13:30 LT. The straight red lines designate linear fits.

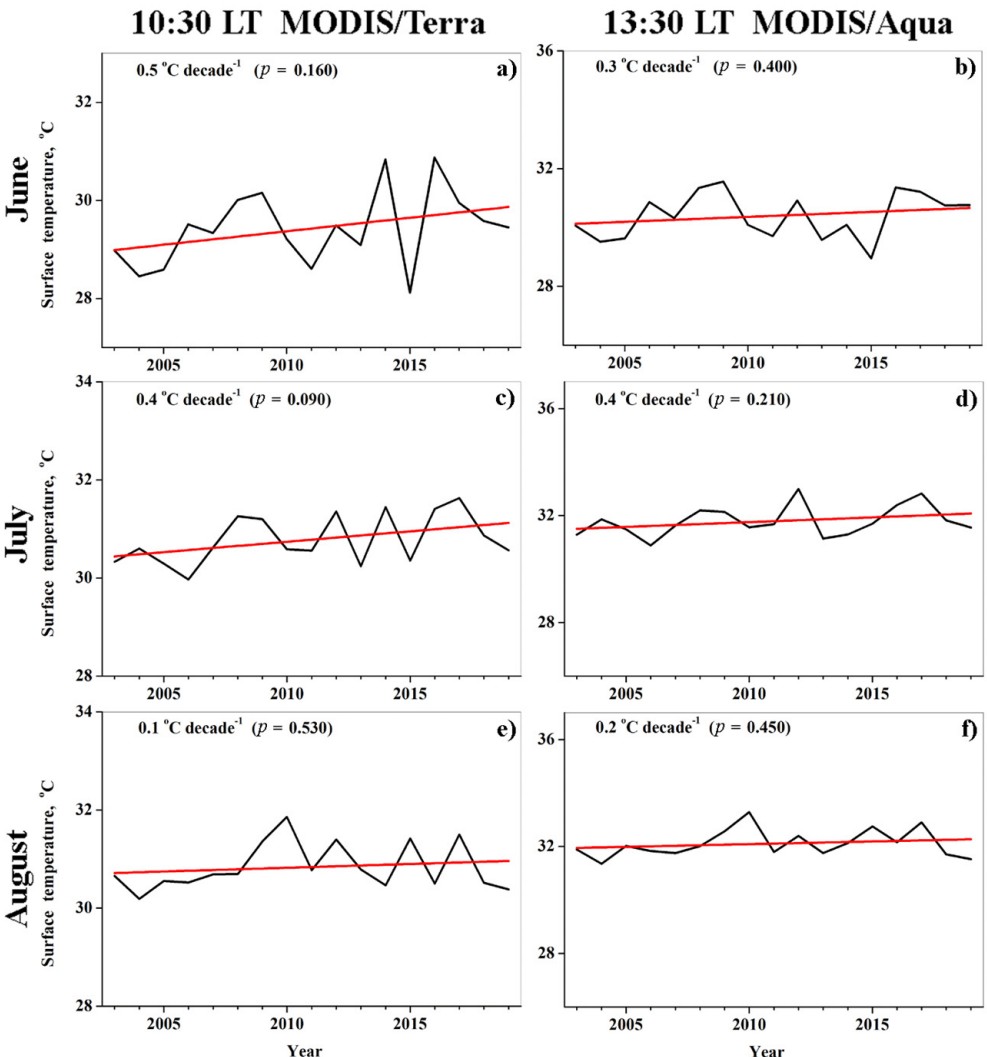

**Figure 12.** Satellite-based MODIS trends in daytime SWT (averaged over the whole Kinneret water area), in the summer months, based on (**left column**) (**a**,**c**,**e**) MODIS/Terra data at 10:30 LT and on (**right column**) (**b**,**d**,**f**) MODIS/Aqua data at 13:30 LT. The top panel (**a**,**b**) corresponds to June, the middle panel (**c**,**d**) corresponds to July, and the bottom panel (**e**,**f**) corresponds to August. The straight red lines designate linear fits.

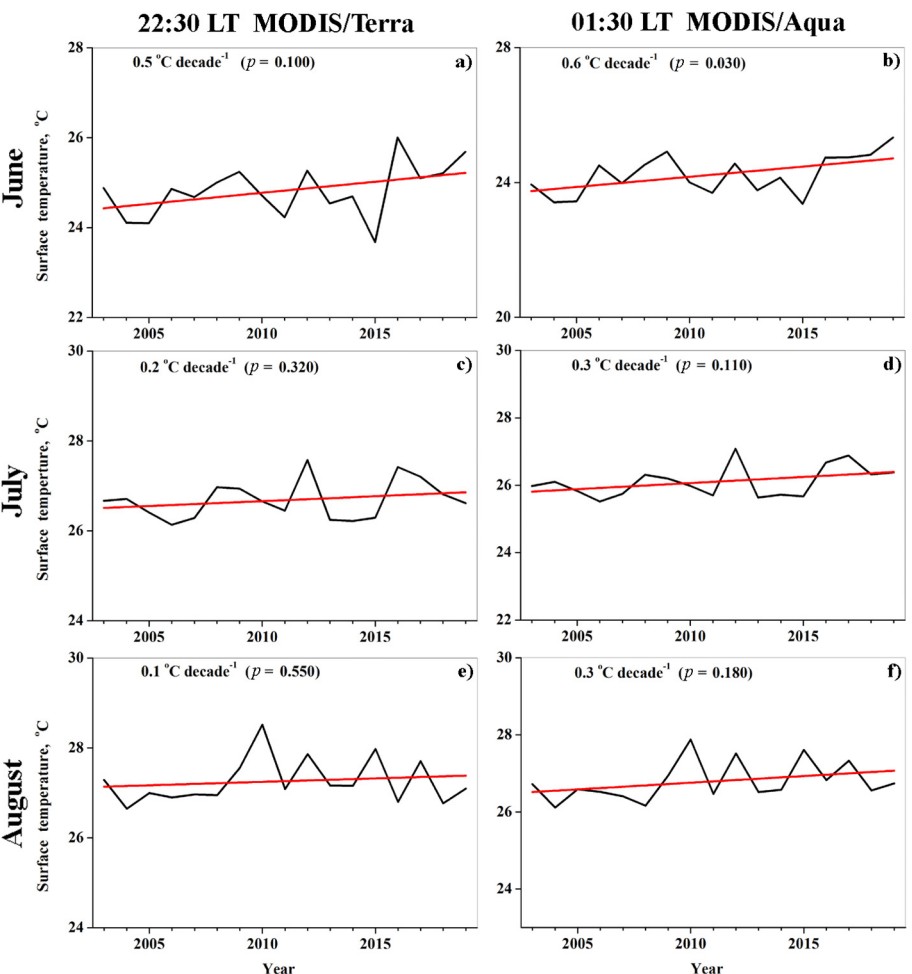

**Figure 13.** Satellite-based MODIS trends in nighttime SWT (averaged over the whole Kinneret water area), in the summer months, based on (**left column**) (**a**,**c**,**e**) MODIS/Terra data at 22:30 LT and on (**right column**) (**b**,**d**,**f**) MODIS/Aqua data at 01:30 LT. The top panel (**a**,**b**) corresponds to June, the middle panel (**c**,**d**) corresponds to July, and the bottom panel (**e**,**f**) corresponds to August. The straight red lines designate linear fits.

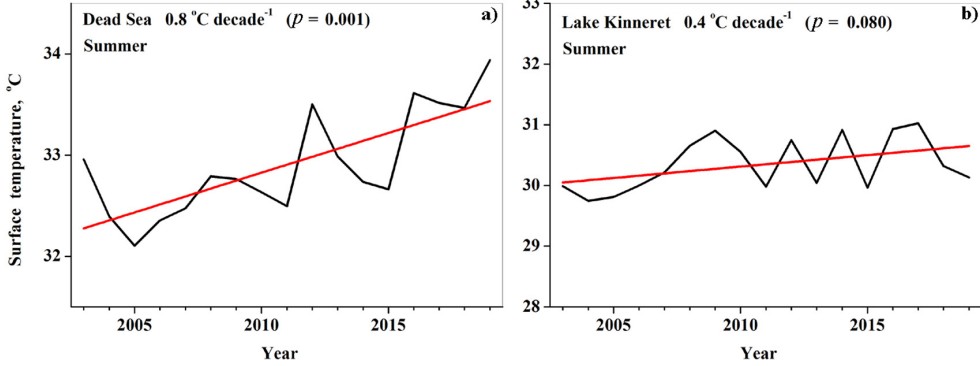

**Figure 14.** Comparison between (**a**) the Dead Sea and (**b**) Lake Kinneret of year-to-year variations of MODIS/Terra surface water temperature in the summer season (June–August), during the 17-year period (2003–2019). The straight red lines designate linear fits.

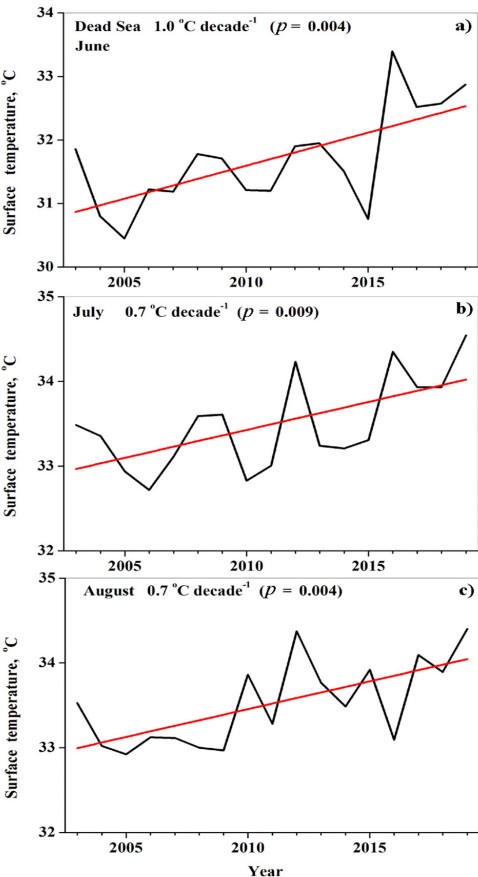

**Figure 15.** Year-to-year variations of satellite MODIS/Terra surface water temperature in the Dead Sea in (**a**) June, (**b**) July, and (**c**) August, during the period 2003–2019. The straight red lines designate linear fits.

## 4. Discussion

The observed increasing atmospheric warming influences both surface water temperature and water temperature below the surface. We found, however, that there was a disagreement between in-situ and satellite measurements with respect to summer surface water temperature trends. In particular, during the study period (2003–2019), in the summer season, between 09:00 LT and 10:00 LT, in-situ measurements of near-surface water temperature showed an increasing trend of 0.7 °C decade$^{-1}$, which reflected the influence of increasing atmospheric warming (Figure 7). During the same period, at 10:30 LT in the summer season, satellite MODIS/Terra high resolution SWT data (measured in the skin layer of 10–20 μm) showed no statistically-significant SWT trends despite the observed increasing atmospheric warming (Figure 11a). Moreover, MODIS/Terra data showed no statistically significant SWT trend in any separate summer month (Figure 12a,c,e). This was observed despite the presence of strong atmospheric warming in each summer month, with the maximal air temperature trend of 1.3 °C decade$^{-1}$ in August (Figure 10a,d,g). In contrast to satellite MODIS/Terra data, in-situ measurements of near-surface water temperature in Lake Kinneret showed increasing trends in each summer month, reflecting the influence of increasing atmospheric warming (Figure 8a–c).

Unlike in-situ measured trends, satellite-based surface water temperature trends are determined not only by increasing atmospheric warming, but also by trends in evaporation. It should be noted that evaporation in Lake Kinneret is maximal in the summer season, particularly in August, in accordance with Rimmer et al. [36]; and Shilo et al. [37]. In summer, the absence of SWT trends in the presence of increasing atmospheric warming is an indication of the influence of steadily increasing evaporation on SWT. Increasing water cooling, due to the above mentioned steadily increasing evaporation, compensated

for increasing heating of surface water by regional atmospheric warming, resulting in statistically insignificant SWT trends. Together with the observed decrease in precipitation [38–40], the above-mentioned increasing evaporation could lead to an essential decrease in Kinneret water levels in the future. The increasing trend in evaporation is in line with model predictions conducted by Rimmer et al. [36] and by La Fuente et al. [41]. According to Rimmer et al. [36], during the period (2015–2060), a combination of high-resolution regional climate models together with a lake evaporation model predicted an increase of 0.10–0.25% in Kinneret evaporation annually: this corresponds to an increase of up to 11% by the year 2060. The predicted increase in evaporation was accompanied by a decrease of 0.8% in annual precipitation corresponding to a decrease of up to 36% by the year 2060. Similar model predictions were obtained by La Fuente et al. [41]. Using ensemble model predictions, they showed an increase of 25% in evaporation and a decrease of 33% in precipitation, resulting in a 58% decline in the water availability of Lake Kinneret, by the end of the 21st century [41].

In-situ measurements of near-surface water temperature were conducted at a specified place near the lake center. We estimated long-term trends in MODIS/Terra SWT at the same place where in-situ measurements were conducted. We found that, in contrast to in-situ measurements, in August, satellite data at the specified place, near the lake center, showed no statistically significant trends in Kinneret SWT (Figure 16a,b). This MODIS/Terra SWT trend at the lake center was in agreement with that of SWT averaged over the whole water area, in August (Figure 12e).

Note that the above-mentioned increasing trend in Lake Kinneret evaporation, in August, was observed in the absence of any trend in wind speed (Figure 16c). Sometimes, however, in hot summer days, synoptic conditions could cause a significant decrease in wind speed over Lake Kinneret. This decrease in wind speed was accompanied by a decrease in evaporation, according to Shilo et al. [37]. It is worth mentioning that, based on available meteorological wind speed data, such phenomena are unable to influence long-term trends in wind speed, and, consequently, these phenomena are not capable of changing increasing trends in evaporation.

In the summer months, during the study period, based on satellite data, we found the absence of statistically significant trends in Kinneret SWT not only in the daytime but also at night (Figure 12a,c,e and Figure 13a,c,e).

For all months during the study period (2003–2019), in-situ measurements showed increasing trends of 0.6 °C decade$^{-1}$ of near surface water temperature (Figure 3b). These trends exceeded the previously published trends: 0.4 °C decade$^{-1}$ during the 50-year period (1970–2019) [12], and 0.28 °C decade$^{-1}$ over the 40-year period (1969–2008) [13]. As a result of this comparison, we can see accelerating trends in Kinneret near-surface temperature. This reflects accelerating regional atmospheric warming (confirmed by available meteorological measurements of air temperature [40,42,43]) and increasing solar radiation (Figure 4f).

It is important to highlight that, in the hypersaline Dead Sea, evaporation is also maximal in the summer months, according to Metzger et al. [44]. However, in contrast to Lake Kinneret, in the Dead Sea, in summer, satellite MODIS/Terra data (1 km × 1 km) showed a strong increasing trend of 0.8 °C decade$^{-1}$ during the same period (2003–2019). The presence of increasing SWT trends in the presence of increasing atmospheric warming is an indication of the absence of steadily increasing evaporation in the Dead Sea. This is supported by a constant drop in Dead Sea water level at the rate of ~1 m/year from year to year during the last 25-year period (1995–2020). As illustrated in Figure 17a, measured yearly data of Dead Sea water levels showed a steady decrease during the period from 1993 to 2020. Our estimated rates of Dead Sea water level drop revealed a constant drop at the rate of ~1 m/year from year to year during the last 25-year period (1995–2020) (Figure 17b). This constant rate of water level drop was observed despite increasing atmospheric warming, indicating the absence of evaporation trends in the hypersaline Dead Sea. This could be explained by increasing surface water salinity in the Dead Sea

skin layer, as a result of increasing evaporation. In its turn, this increasing surface water salinity suppresses further increases in evaporation. As a result, there was no acceleration in Dead Sea water level drop in the presence of an increasing SWT trend of 0.8 °C decade$^{-1}$. We consider that this is a characteristic feature of the hypersaline Dead Sea, which is not present in the fresh-water Lake Kinneret.

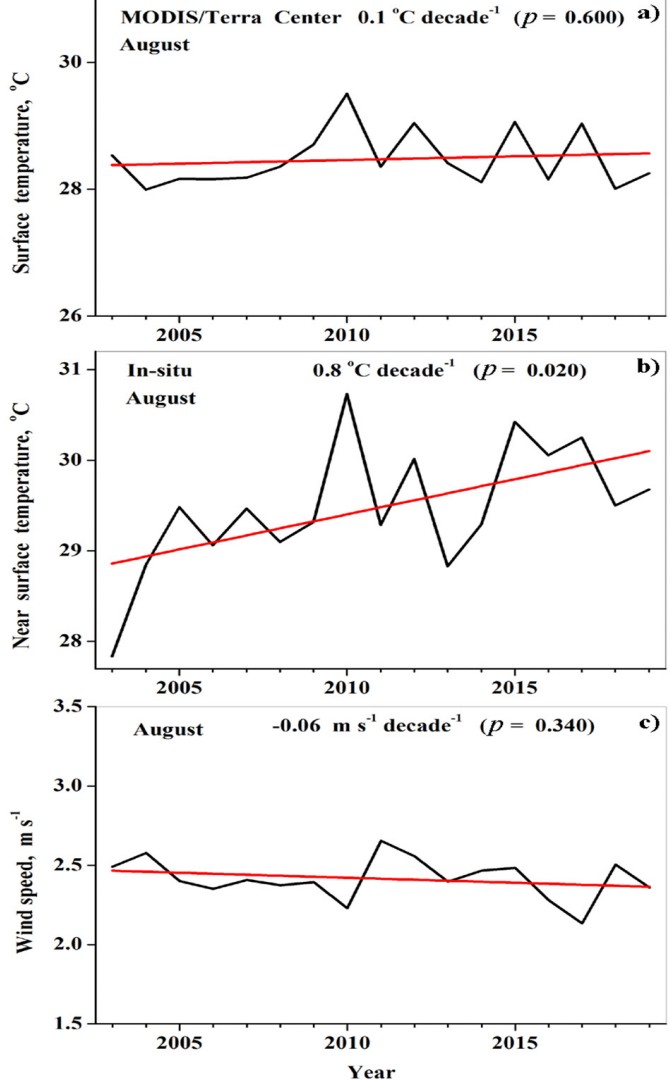

**Figure 16.** Year-to-year variations of (**a**) MODIS/Terra SWT at a specified place near the Kinneret center, (**b**) in-situ measurements of near surface water temperature at the same place, and (**c**) wind speed taken at the Zemah meteorological station, in August, during the 17-year period (2003–2019). The straight red lines designate linear fits.

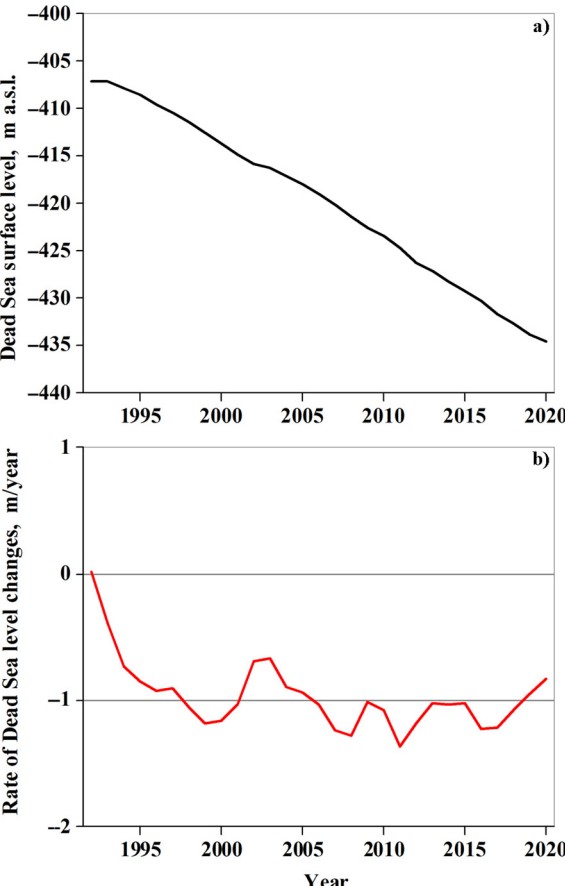

**Figure 17.** (**a**) Yearly data of Dead Sea water levels based on available measurements from 1992 to 2020; (**b**) the rate of Dead Sea water level drop from year to year.

## 5. Conclusions

In this study we investigated trends in Kinneret surface water temperature (SWT) using MODIS 1 km × 1 km resolution records, on board Terra and Aqua satellites, during the period from 2003 to 2019.

We found that, in summer when evaporation is maximal, despite the presence of increasing atmospheric warming, satellite data revealed the absence of SWT trends in fresh-water Lake Kinneret. The absence of SWT trends in the presence of increasing atmospheric warming is an indication of the influence of steadily increasing evaporation on SWT. Increasing water cooling, due to the above mentioned steadily increasing evaporation, compensated for increasing heating of surface water by regional atmospheric warming. This resulted in the obtained statistically insignificant SWT trends. Together with the observed decrease in precipitation in the rainy season (from October to early May) [38–40], the above-mentioned increasing evaporation could lead to an essential decrease in Kinneret water levels in the future. This increasing evaporation is in line with model predictions. Rimmer et al. [36] predicted an increase of 11% in Kinneret evaporation and a decrease of 36% in precipitation by the year of 2060, using a combination of high-resolution regional climate models together with a lake evaporation model. Similarly, ensemble model predictions by La Fuente et al. [41] showed an increase of 25% in evaporation and a decrease of 33% in precipitation: this resulted in a 58% decline in the water availability of Lake Kinneret, by the end of the 21st century. As Lake Kinneret supplies fresh water to both Israel and Jordan, the predicted desiccation of the lake creates water supply problems for both countries.

In contrast to fresh-water Lake Kinneret, in the hypersaline Dead Sea, located ~100 km apart, MODIS records showed an increasing statistically significant trend of 0.8 °C decade$^{-1}$ in summer SWT, during the same study period. The presence of increasing SWT trends in

the presence of increasing atmospheric warming is an indication of the absence of steadily increasing evaporation in the Dead Sea. This is supported by a constant drop in Dead Sea water level at the rate of ~1 m/year from year to year during the last 25-year period (1995–2020) (Figure 17b). This could be explained by increasing surface water salinity in the Dead Sea skin layer, as a result of increasing evaporation. In its turn, this increasing surface water salinity suppresses further increases in evaporation. As a result, there was no acceleration in Dead Sea water level drop in the presence of an increasing SWT trend of 0.8 °C decade$^{-1}$. We consider that this is a characteristic feature of the hypersaline Dead Sea, which is not present in the fresh-water Lake Kinneret.

During the study period (2003–2019), in summer, in contrast to satellite data, in-situ measurements of near-surface water temperature in Lake Kinneret showed an increasing trend of 0.7 °C decade$^{-1}$. This trend reflected the presence of accelerating atmospheric warming and increasing solar radiation. During the same period, for all months, in-situ measurements of near-surface water temperature in Lake Kinneret showed an increasing trend of 0.6 °C decade$^{-1}$. These trends exceeded the previously published trend of 0.4 °C decade$^{-1}$ during the 50-year period (1970–2019) [12], and that of 0.28 °C decade$^{-1}$ over the 40-year period (1969–2008) [13]. These previously published trends were also based on in-situ measurements in Lake Kinneret. Consequently, in-situ measurements, conducted during the three above mentioned periods, showed the accelerating trend in Kinneret near-surface water temperature. It is important to highlight that the above-mentioned accelerated trends in Kinneret near-surface water temperature have potential consequences for a broad range of physical and ecological factors such as thermal structure; lake productivity and ecosystems; as well as lake desiccation.

**Author Contributions:** All co-authors equally contributed to the writing of the current research article: P.K., B.S., Y.L., P.A.; shipboard measurements: Y.L. All authors have read and agreed to the published version of the manuscript.

**Funding:** This research received no external funding.

**Institutional Review Board Statement:** Not applicable.

**Informed Consent Statement:** Not applicable.

**Data Availability Statement:** The research data used in the current study are publicly available [14,15,24].

**Acknowledgments:** We thank the MODIS teams that produced the data used in this study: Collections-6 of MODIS/Terra MOD11A1 and MODIS/Aqua MYD11A1 Level 3 LST data products. We thank the Israel Meteorological Service for measurements of 2-m air temperature, surface solar radiation, and wind speed, during the 17-year study period, taken at the Zemah meteorological station. Credit for the data of Lake Kinneret and Dead Sea water levels is given to Israel Hydrological Service. Kinneret water temperature data presented in this work are part of the Israeli Water Authority monitoring program of Lake Kinneret, which is performed by the Yigal Alon Kinneret Limnological Laboratory, Israel Oceanographic and Limnological Research. We thank all reviewers for their helpful comments.

**Conflicts of Interest:** The authors declare no conflict of interest.

## Appendix A

**Table A1.** The slope ($\alpha$) with its standard error (SE) of the obtained linear fit of deseasonalized monthly anomalies of near-surface water temperature (NSWT), water temperature at a depth of 1 m (WT-1m), and of 2 m (WT-2m), based on in-situ observations between 9 LT–10 LT, during the study period (2003–2019). The decision based on the Shapiro-Wilk normality test for residuals (S-W test) and the significance level (*p*) is also displayed. If the *p* value was too high as compared with the 0.05 significance level, the obtained linear fit was considered as statistically insignificant. In addition, slopes were obtained for year-to-year variations of MSWT in the summer season and in each summer month separately.

| Temperature [°C] | $\alpha \pm$ SE [°C Decade$^{-1}$] | S–W Test | $p$ |
|---|---|---|---|
| NSWT (all months) | $0.6 \pm 0.1$ | Normal | 0.001 |
| NSWT (Summer) | $0.7 \pm 0.1$ | Normal | 0.001 |
| NSWT (June) | $0.7 \pm 0.2$ | Normal | 0.020 |
| NSWT (July) | $0.7 \pm 0.2$ | Normal | 0.010 |
| NSWT (August) | $0.8 \pm 0.2$ | Normal | 0.020 |
| WT-1m (all months) | $0.6 \pm 0.1$ | Normal | 0.001 |
| WT-2m (all months) | $0.6 \pm 0.1$ | Normal | 0.001 |

**Table A2.** The slope ($\alpha$) with its standard error (SE) of the obtained linear fit of deseasonalized monthly anomalies of air temperature (Tair), wind speed (WS), and solar radiation (SR) measured at the Zemah meteorological station (9 LT–10 LT), during the study period. The decision based on the Shapiro-Wilk normality test for residuals (S-W test) and the significance level (*p*) is also displayed. In addition, slopes were obtained for year-to-year variations of Tair/WS/RS in the summer season and in each summer month separately.

| Air Temperature [°C] | $\alpha \pm$ SE [°C decade$^{-1}$] | S–W Test | $p$ |
|---|---|---|---|
| Tair (all months) | $1.0 \pm 0.1$ | Normal | 0.001 |
| Tair (Summer) | $1.1 \pm 0.2$ | Normal | 0.001 |
| Tair (June) | $1.2 \pm 0.2$ | Normal | 0.001 |
| Tair (July) | $0.9 \pm 0.3$ | Normal | 0.008 |
| Tair (August) | $1.3 \pm 0.4$ | Normal | 0.006 |
| **Wind Speed (m s$^{-1}$)** | $\alpha \pm$ **SE [m s$^{-1}$ decade$^{-1}$]** | **S–W Test** | $p$ |
| WS (all months) | $0.01 \pm 0.03$ | Normal | Not significant (0.680) |
| WS (Summer) | $-0.02 \pm 0.05$ | Normal | Not significant (0.720) |
| WS (June) | $-0.02 \pm 0.10$ | Normal | Not significant (0.830) |
| WS (July) | $0.03 \pm 0.06$ | Normal | Not significant (0.640) |
| WS (August) | $-0.06 \pm 0.06$ | Normal | Not significant (0.450) |
| **Solar Radiation (W m$^{-2}$)** | $\alpha \pm$ **SE [W m$^{-2}$ decade$^{-1}$]** | **S–W Test** | $p$ |
| SR (all months) | $20 \pm 4$ | Normal | 0.001 |
| SR (Summer) | $16 \pm 8$ | Normal | 0.050 |
| SR (June) | $9 \pm 14$ | Normal | Not significant (0.540) |
| SR (July) | $22 \pm 11$ | Normal | 0.050 |
| SR (August) | $16 \pm 13$ | Normal | Not significant (0.250) |

**Table A3.** The slope ($\alpha$) with its standard error (SE) of the obtained linear fit of deseasonalized monthly anomalies of surface water temperature (SWT) of Lake Kinneret based on satellite data MODIS/Terra and MODIS/Aqua, during the study period (2003–2019). The decision based on the Shapiro-Wilk normality test for residuals (S-W test) and the significance level (*p*) is also displayed. In addition, slopes were obtained for year-to-year variations of MODIS SWT in the summer season and in each summer month separately, both in Lake Kinneret and in the Dead Sea.

| Temperature [°C] | $\alpha \pm$ SE [°C Decade$^{-1}$] | S–W Test | *p* |
|---|---|---|---|
| Kinneret SWT Terra 10:30 LT (all months) | $0.5 \pm 0.1$ | Normal | 0.001 |
| Kinneret SWT Terra 10:30 LT (Summer) | $0.4 \pm 0.2$ | Normal | Not significant (0.080) |
| Kinneret SWT Terra 10:30 LT (June) | $0.5 \pm 0.3$ | Normal | Not significant (0.160) |
| Kinneret SWT Terra 10:30 LT (July) | $0.4 \pm 0.2$ | Normal | Not significant (0.090) |
| Kinneret SWT Terra 10:30 LT (August) | $0.1 \pm 0.2$ | Normal | Not significant (0.530) |
| Kinneret-Center SWT Terra 10:30 LT (August) | $0.1 \pm 0.2$ | Normal | Not significant (0.530) |
| Kinneret SWT Terra 22:30 LT (June) | $0.5 \pm 0.3$ | Normal | Not significant (0.100) |
| Kinneret SWT Terra 22:30 LT (July) | $0.2 \pm 0.2$ | Normal | Not significant (0.320) |
| Kinneret SWT Terra 22:30 LT (August) | $0.1 \pm 0.2$ | Normal | Not significant (0.550) |
| Kinneret SWT Aqua 13:30 LT (all months) | $0.4 \pm 0.1$ | Normal | 0.001 |
| Kinneret SWT Aqua 13:30 LT (Summer) | $0.3 \pm 0.2$ | Normal | Not significant (0.210) |
| Kinneret SWT Aqua 13:30 LT (June) | $0.3 \pm 0.4$ | Normal | Not significant (0.400) |
| Kinneret SWT Aqua 13:30 LT (July) | $0.4 \pm 0.3$ | Normal | Not significant (0.210) |
| Kinneret SWT Aqua 13:30 LT (August) | $0.2 \pm 0.2$ | Normal | Not significant (0.450) |
| Kinneret SWT Aqua 01:30 LT (June) | $0.6 \pm 0.3$ | Normal | 0.030 |
| Kinneret SWT Aqua 01:30 LT (July) | $0.3 \pm 0.2$ | Normal | Not significant (0.110) |
| Kinneret SWT Aqua 01:30 LT (August) | $0.3 \pm 0.2$ | Normal | Not significant (0.180) |
| Dead Sea SWT Terra 10:30 LT (Summer) | $0.8 \pm 0.2$ | Normal | 0.001 |
| Dead Sea SWT Terra 10:30 LT (June) | $1.0 \pm 0.3$ | Normal | 0.004 |
| Dead Sea SWT Terra 10:30 LT (July) | $0.7 \pm 0.2$ | Normal | 0.009 |
| Dead Sea SWT Terra 10:30 LT (August) | $0.7 \pm 0.2$ | Normal | 0.004 |

## Appendix B

**Table A4.** Yearly data of the Kinneret water levels (m a.s.l.) from 1935 to 2020.

| Year | Water Level | Year | Water Level | Year | Water Level |
|------|-------------|------|-------------|------|-------------|
| 1935 | −209.85 | 1965 | −211.20 | 1995 | −209.84 |
| 1936 | −209.61 | 1966 | −210.36 | 1996 | −210.44 |
| 1937 | −209.43 | 1967 | −209.19 | 1997 | −211.12 |
| 1938 | −209.34 | 1968 | −209.30 | 1998 | −211.49 |
| 1939 | −209.67 | 1969 | −209.01 | 1999 | −212.42 |
| 1940 | −209.80 | 1970 | −209.46 | 2000 | −212.92 |
| 1941 | −209.91 | 1971 | −209.58 | 2001 | −213.87 |
| 1942 | −210.00 | 1972 | −209.73 | 2002 | −213.91 |
| 1943 | −210.27 | 1973 | −210.72 | 2003 | −210.98 |
| 1944 | −210.23 | 1974 | −210.44 | 2004 | −209.83 |
| 1945 | −210.21 | 1975 | −210.88 | 2005 | −210.60 |
| 1946 | −210.59 | 1976 | −210.64 | 2006 | −211.38 |
| 1947 | −210.44 | 1977 | −210.16 | 2007 | −211.85 |
| 1948 | −210.26 | 1978 | −209.53 | 2008 | −213.07 |
| 1949 | −210.17 | 1979 | −210.73 | 2009 | −213.93 |
| 1950 | −211.01 | 1980 | −209.80 | 2010 | −213.35 |
| 1951 | −210.24 | 1981 | −209.50 | 2011 | −213.15 |
| 1952 | −210.21 | 1982 | −210.51 | 2012 | −212.08 |
| 1953 | −210.45 | 1983 | −210.10 | 2013 | −210.62 |
| 1954 | −210.74 | 1984 | −210.09 | 2014 | −211.85 |
| 1955 | −210.79 | 1985 | −210.50 | 2015 | −212.42 |
| 1956 | −210.37 | 1986 | −211.65 | 2016 | −213.02 |
| 1957 | −210.69 | 1987 | −210.35 | 2017 | −213.59 |
| 1958 | −210.37 | 1988 | −209.63 | 2018 | −213.98 |
| 1959 | −210.74 | 1989 | −211.04 | 2019 | −211.96 |
| 1960 | −210.84 | 1990 | −212.02 | 2020 | −209.61 |
| 1961 | −210.75 | 1991 | −212.40 | | |
| 1962 | −210.81 | 1992 | −209.41 | | |
| 1963 | −210.54 | 1993 | −209.17 | | |
| 1964 | −210.70 | 1994 | −209.78 | | |

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
