# Peer review of "Absence of Surface Water Temperature Trends in Lake Kinneret despite Present Atmospheric Warming: Comparisons with Dead Sea Trends"

_remotesensing, doi:10.3390/rs13173461_

Round 1
Reviewer 1 Report
The authors utilized lake surface temperature data from the MODIS instrument and weekly in situ measurements to compare the decadal trends of skin and bulk surface water temperatures in Lake Kinnereth. Additionally, the skin temperature trends were estimated for the neighbouring Dead Sea from the MODIS data. The results demostrated a weaker and less insignificant trend in the summer skin temperatures of Lake Kinnereth as compared to the bulk water temperatures and to the Dead Sea skin temperatures. The authors ascribe this result to the effect of increased summer evaporation from the freshwater surface, which dampens the skin temperature growth driven by atmospheric warming. The finding reveals a potentially important facet of the freshwater lake response the global change, as well as contributes to a correct interpretation of lake surface temperatures derived from remote sensing. By this, the study is of interest for limnologists, boundary layer meteorologists, and remote sensing scientists. The methods applied in the study are generally adequate, while an improvement of robustness is possible, especially with regard to the trend significance estimation for relatively short data series. The manuscript structure also has a room for improvement to better distribute the information between results and discussion; the presentation is sometimes imbalanced, with some unnecessary information counterparted by too scarce or missing background information at other places. The language and style are generally acceptable; more concise presentation, removal of repetitions, and refinement of style are advised. The detailed remarks are below.
-- Introduction, Lines 48-66: How the detailed description of the diurnal wind variability relates to the subject of the study (monthly lake surface temperatures)? Clarify or shortren accordingly, retaining the essential information only.
-- Introduction, Lines 70-73: In contrast to Lake Kinnereth, no background information is presented on the Dead Sea. What is the wind pattern over the lake, is it the same as over Lake Kinnereth? Have any climatic temperature trends been previously reported for the Dead Sea? How comparable are both water bodies in terms of the surface area, morphometry, hydrological regime, and mean temperatures? Please, extend.
-- Materials and Methods, Line 99: "the slope of the linear fit". Was the trend estimated using the ordinary least squares (simple linear regression)? How robust is the method for the short (17 points) and potentially autocorrelated data series? It is advised to additionally apply non-parametric trend estimators, like the Mann-Kendall test and the Sen's slope estimator, and compare the outcomes against the simple liear regression results. It is a key point of the entire analysis, since the absence of a significant trend in summer months is the major result of the study. Do alternative trend estimators support the trend absence?
-- Line 57 and everywhere: "7 LT" use the standard daytime writing, "07:00 LT"
-- Line 113 and everywhere: "aboard ship" replace with "shipboard"
-- Line 127: "available measurements from 1935 to 2020" provide a reference to the data source
- Results, Line 143 and everywhere: "middle of 1960s" replace with "mid-1960s"
-- Line 152: clarifiy, what does "lake was completely full" mean
-- Lines 152-153: "the dam was open" - add an information on dam operation to the lake description in Introduction
-- Lines 235-236: "To this end..." the sentence is a repetition of Methods. Remove
-- Line 253: "This is evidence..." This statement requires detailed explanation. Why is it an evidence of the climate change effect? Which climate-related mechanisms may be responsible for the effect? Why other potential cuses can be discarded? Extend (or remove the sentence completely)
-- Lines 322-331: The paragraph can be shortened to a half without loosing any essential information.
-- Discussion, Lines 404-413: The paragraph mostly repeates the previous information. Shorten significantly, retaining only the essential part.
-- Line 443 onwards: Section 4.3 should be divided in Results and Discussion. Present the results from the Dead Sea in the "Results" section, and discuss the similarities and differences of outcomes from Lake Kinnereth and the Dead Sea in the "Discussion" section.
-- See also suggestions on style in the annotated pdf file (they are not exhaustive, reworking of the style towards more concise presentation across the entire ms is advised)

Author Response
See our answers to the reviewer-1's comments in the attached file.

Reviewer 2 Report
The manuscript use the long time series of the in-situ SWT and MODIS SWT to analyze the SWT change tendency of the Lake Kinneret Dead Sea. It is a interesting topic un the global climate background.
In the method section, the authors had better give the formulars of the statistical and computational methods.
How to calculate the Figure2 (b)?
How about the accuracy of the MODIS LST products in you study area?
Line 221-225, The relationship among the surface water temperature, air temperature and surface solar radiationcan be analyzed using correlation analysis.
Line 234, The MODIS SWT were averaged over the specified Kinneret water area. I think that there is no comparison between MODIS SWT with the in-situ SWT. You had better compare the in-situ SWT with MODIS SWT of the same pixel.
Author Response
See our answers to the reviewer-2' comments in the attached file.

Round 2
Reviewer 2 Report
The authors had revised the manuscript according to my commments.